# DySL-VLA: Efficient Vision-Language-Action Model Inference via Dynamic-Static Layer-Skipping for Robot Manipulation

## Abstract

Vision-Language-Action (VLA) models have shown remarkable success in robotic tasks like manipulation by fusing a language model's reasoning with a vision model's 3D understanding. However, their high computational cost remains a major obstacle for real-world applications that require real-time performance. We observe that the actions within a task have varying levels of importance: critical steps demand high precision, while less important ones can tolerate more variance. Leveraging this insight, we propose DySL-VLA, a novel framework that addresses computational cost by dynamically skipping VLA layers based on each action's importance. DySL-VLA categorizes its layers into two types: informative layers, which are consistently executed, and incremental layers, which can be selectively skipped. To intelligently skip layers without sacrificing accuracy, we invent a prior-post skipping guidance mechanism to determine when to initiate layer-skipping. We also propose a skip-aware two-stage knowledge distillation algorithm to efficiently train a standard VLA into a DySL-VLA. Our comprehensive experiments indicate that DySL-VLA surpasses the state of the art, achieving a 2.1% improvement in success length over Deer-VLA (NeurIPS'24) on the Calvin dataset, while simultaneously reducing trainable parameters by a factor of 85.7 and providing a $3.75\times$ speedup relative to the RoboFlamingo baseline at iso-accuracy. Our code is available on Anonymous Github.

## 1 Introduction

Inspired by the success of Vision-Language Models (VLMs) Alayrac et al. (2022); An et al. (2024); Li et al. (2022); Luo et al. (2024); Li et al. (2023a), Vision-Language-Action (VLA) models have emerged Brohan et al. (2023; 2022); Wen et al. (2025); Kim et al. (2025); Wang et al. (2025c), enabling a promising paradigm for end-to-end robotic control. By tokenizing robot control signals, these models take images as environmental observations and language instructions as the task goal, then generate the next control command for the robot to fulfill the task Li et al. (2023b); Kim et al. (2024). Leveraging the vast, internet-scale knowledge embedded within VLMs Brohan et al. (2023); Kim et al. (2024), VLA models have already shown remarkable generalization capabilities in complex robotic tasks like manipulation Fan et al. (2025); Liu et al. (2024a).

However, deploying VLA models on real-world robots poses a significant challenge. Their immense computational demands lead to high latency and power consumption Yue et al. (2024); Zhang et al. (2025); Li et al. (2025); Xu et al. (2025b); Wang et al. (2025b), which conflict with the limited resources and battery capacity of most robotic platforms Karumbunathan (2022); Valladares et al. (2021). Consequently, existing VLA systems, like RT-2 (1-3 Hz) Brohan et al. (2023) and OpenVLA (3-5 Hz) Kim et al. (2024), have slow action generation speeds compared to the high-frequency low-level control required for real-time physical interaction (20-50+ Hz) Kim et al. (2025).

While existing VLA acceleration methods, such as quantization Park et al. (2024); Chen & Li (2025), pruning Zhang et al. (2024b); Chen et al. (2024), and knowledge distillation Zhang et al. (2025); Chen & Li (2025), have not fully solved this problem, they often overlook a crucial insight: in robot manipulation, the importance of different actions is not equal. For instance, the act of grasping or releasing an object is far more critical to a task's success than the preparatory pre-grasp movements,

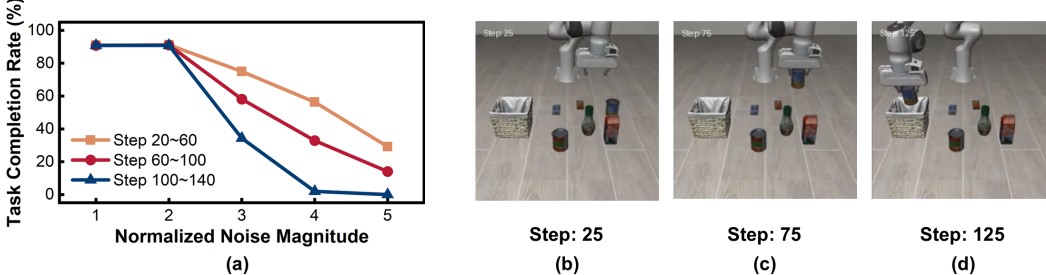

Figure 1: Different actions in robot manipulation have different importance. We show an example when the robot is performing task "Grasp the black cup and drop it into basket". (a) shows the task completion rates when adding noise with different magnitudes to VLA model weights at different action steps. When adding noise at important action steps, the task completion rate drops faster as noise magnitude increases. We sample 50 times on each noise magnitude for each step range. We show the robot status at (b) step 25, (c) step 75, and (d) step 125 when using the origin VLA model.

which is also shown in Figure 1. By applying a uniform approach to all action predictions, these methods miss key opportunities for acceleration on less important actions and, therefore, offer limited speedup. Similarly, early exit methods Yue et al. (2024); Song et al. (2025a) attempt to take advantage of this feature by dynamically adjusting the computational load, but they risk discarding crucial information by exiting before the final layers are fully processed. This trade-off can compromise the model's overall accuracy and effectiveness Zhang et al. (2025).

To address the high latency and computational demands of VLA models, we propose DySL-VLA, a method that dynamically skips unnecessary layers during inference. Our approach is based on a key finding: not all VLA layers contribute equally to action prediction. Specifically, we observed that activation distributions change significantly after certain "informative" layers. Our dynamic-static layer skipping method leverages this insight by statically keeping the most critical layers while dynamically skipping others. We also found that the success of a manipulation task is highly sensitive to the accuracy of a few key actions. To account for this and ensure training convergence, we introduce a prior-post skipping guidance and a skip-aware two-stage knowledge distillation method. We summarize our contributions as follows:

- We conduct comprehensive analysis of layer-wise performance and action importance variations in VLA action prediction.

- We propose DySL-VLA to accelerate VLA inference via dynamic-static layer skipping. We also propose prior-post skipping guidance and skip-aware two-stage knowledge distillation to ensure the correctness of important actions and improve training convergence.

- Extensive experiments show that DySL-VLA shows 3.75× latency reduction compared with RoboFlamingo, and 2.1% average successful length improvement compared with DeeR-VLA, with 85.7× trainable parameters and 13.7× training steps reduction.

## 2 BACKGROUND

**Vision-language-action Model.** Numerous studies have investigated instructing robots by natural language Driess et al. (2023); Sun et al. (2024). Among them, VLA models are fine-tuned from pretrained VLMs to increase generalization and conduct robot control in an end-to-end way Black et al. (2024); Zhang et al. (2024a), which shows good performance and becomes the mainstream. As shown in Figure 2, in each frame, VLA model predicts action given the current image observation and language instruction Li et al. (2023b); Kim

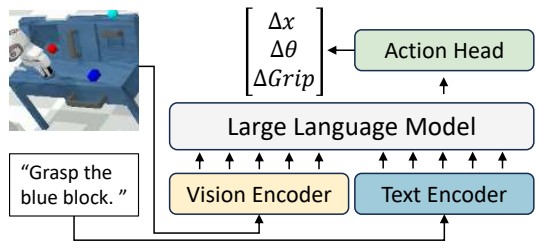

Figure 2: VLA model architecture.

et al. (2024). Though achieving high performance, VLA model shows long real-time latency and low control frequency Wen et al. (2025); Song et al. (2025b), which comes from the high

Table 1: Comparison between different methods.

| Method | Approach | Action Importance Aware | Specialized Kernel Free | Training Modules |
|---|---|---|---|---|
| SparseVLM Zhang et al. (2024b) | Pruning | ✗ | ✗ | – |
| FastV Chen et al. (2024) | Pruning | ✗ | ✗ | – |
| QAIL Park et al. (2024) | Quantization | ✗ | ✗ | LLM Backbone |
| MoLe-VLA Zhang et al. (2025) | Mixture-of-Layers | ✗ | ✓ | LLM Backbone |
| DeeR-VLA Yue et al. (2024) | Early-exit | ✓ | ✓ | LLM Backbone & Action Heads |
| DySL-VLA | Dynamic-static Layer Skipping | ✓ | ✓ | Light-weight Adapters & Skipping Controllers |

computation cost of the LLM backbone. In this paper, we mainly focus on inference acceleration of the LLM backbone of the VLA models, which accounts for most of the parameters and inference latency (84.3% for OpenVLA Kim et al. (2024) and 75.4% for OpenVLA-oft Kim et al. (2025)).

**Efficient Model Inference.** Existing VLA acceleration works use pruning Yang et al. (2025); Wang et al. (2025a); Zhang et al. (2024b), quantization Park et al. (2024); Lin et al. (2024), and mixture-of-layers Zhang et al. (2025) to accelerate VLA models, while these methods ignore the importance difference of each action and allocate an equal amount of computation to each prediction. This wastes the acceleration opportunities on unimportant actions, thus causing a lower acceleration rate. In addition, methods such as quantization and pruning require specialized kernels, which increase the difficulty of deployment. Early-exit methods Teerapittayanon et al. (2016); Xu et al. (2025a); Rahmath P et al. (2024) halting forward propagation at a certain layer based on intermediate predictions, which can dynamically apply computation on different actions. But skipping all final layers results in a significant loss of information. To solve this, DeeR-VLA Yue et al. (2024) largely trains the LLM backbone and multiple action heads to recover model performance. However, this will introduce high computation and memory costs in the training stage. The large-scale fine-tuning on specific scenarios may also break the generalization ability of VLA models. In addition, its exit decision mechanism also introduces non-negligible extra inference costs.

Compared with existing works, our method adaptively applies more computation to important actions using layer skipping methods. We systematically examine the role of each layer and use dynamic-static layer skipping to reduce information loss after skipping. We use pre-skip prediction and post-skip verification to ensure correct skipping decisions. For training efficiency, we only train light-weight skipping controllers and adapters instead of the LLM backbone. The comparison of different methods for VLA acceleration is shown in Table 1.

## 3 Efficient VLA Inference via Dynamic-Static Layer-Skipping

### 3.1 Observation and Overview

To achieve high acceleration rate and model performance after layer skipping, there are two questions to answer. 1) When should we conduct layer skipping? 2) Which layer should we skip to if we conduct layer skipping? To answer the questions, we conduct the following observations.

**Observation 1: the importance across different VLA layers varies a lot, while skipping informative layers may cause low model performance.** When deciding which layers we should skip to, existing layer skipping works Yue et al. (2024); Luo et al. (2025); Raposo et al. (2024); Fan et al. (2024) either empirically skip with the same interval or directly skip all the final layers. However, these strategies do not consider the different importance of VLA layers. We evaluate the amount of information contained in each layer of VLA models by calculating the average cosine similarity of each layer's output activation. As shown in Figure 3 (a) and (b), we find that some VLA layers significantly change the activation distribution compared to other layers. As shown in Figure 3 (c) and (d), skipping these informative layers will introduce significant performance drops.

**Observation 2: VLA systems show high sensitivity on important actions, and more restrictions are needed to decide skipping positions.** To decide when to skip layers, existing works Jiang et al. (2024); Luo et al. (2025); Raposo et al. (2024) use skipping controllers (usually feedforward networks) before LLM layers to predict skipping probability, and conduct layer skipping if the probability exceeds a threshold. However, directly transferring this method to VLA models shows

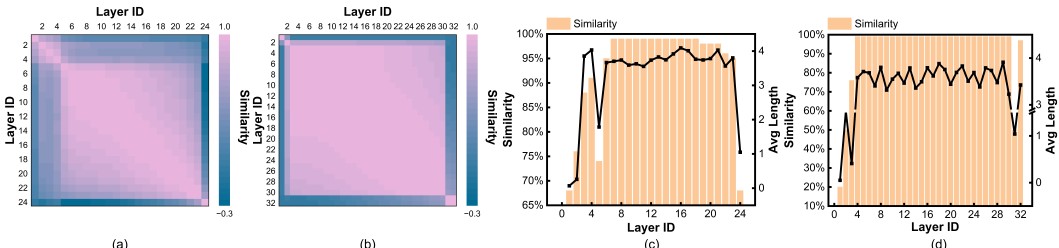

Figure 3: The average cosine similarity between the output activations of different VLA layers for (a) RoboFlamingo-3B and (b) RoboFlamingo-9B. The similarity between the input and output activations of each layer and the model performance when skipping each VLA layer in a zero-shot manner for (c) RoboFlamingo-3B and (d) RoboFlamingo-9B.

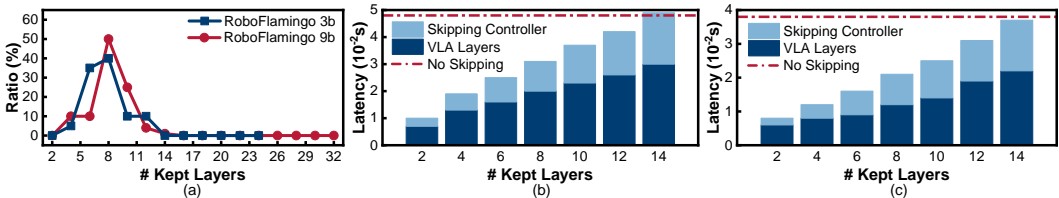

Figure 4: (a) The ratio of different numbers of kept layers in VLA model inference when only using skipping controllers. The inference latency for different numbers of kept layers using (b) RoboFlamingo-3B in FP32 and (c) RoboFlamingo-9B in FP16.

low accuracy. This is because even small errors caused by layer skipping on important actions may cause task failure, which is shown in Figure 1. Only based on the current activation, it is hard for the skipping controller to fully understand the action's importance. As shown in Figure 4 (a), when only using skipping controllers, the number of kept layers is concentrated between 5 to 12 and does not exceed 14, which is not enough for important action predictions. So, when considering when to skip layers, more restrictions are needed to keep more layers for important actions.

In addition, the skipping controllers will introduce non-negligible extra inference latency, which comes from the serial nature of the inference of skipping controllers and the VLA layers Xu et al. (2025a). As shown in Figure 4 (b) and (c), when the skipping mechanism is introduced before each layer and half of the layers are activated, the mechanism will gain little latency reduction compared with the baseline model. New strategies are needed to improve the efficiency of controller inference.

**DySL-VLA overview.** Based on these observations, we propose DySL-VLA, accelerating VLA models by dynamically skipping unnecessary layers according to action importance. To achieve low information loss and high speedup, we propose dynamic-static layer skipping to statically keep the informative layers and dynamically skip unnecessary layers (Section 3.2). To keep enough layers for important action, we propose pre-skip prediction and post-skip verification to guide the skipping decision (Section 3.3). Finally, we propose skip-aware two-stage knowledge distillation to conduct low-cost training and improve training convergence (Section 3.4).

### 3.2 DYNAMIC-STATIC LAYER SKIPPING

**Problems of existing layer skipping works.** Existing layer skipping works do not consider the different importance of VLA layers, thus showing sub-optimal results. As shown in Figure 5 (b) and (c), early exit methods either need to train multiple action heads Yue et al. (2024); Liu et al. (2024b) or introduce adapters Ji et al. (2023) to fit the final action head. However, these methods skip all final layers, some of which are informative, thus showing lower performance. Luo et al. (2025); Jiang et al. (2024) only skip one layer each time, as shown in Figure 5 (d). Although showing reasonable performance, this fine-grained skipping mechanism shows low acceleration rate because the skipping controller and adapter introduce extra inference cost.

To solve these problems, we propose a dynamic-static layer skipping mechanism, as shown in Figure 5 (e). As discussed in Section 3.1, some informative layers in VLA models significantly change the activation distribution. We define them as *static layers* and statically keep these layers in model inference. At the same time, although other layers contain less information, we find that directly

Figure 5: The inference mode of (a) original VLA model, (b) using early exit with multiple action heads, (c) using early exit with adapters, (d) using traditional layer skipping methods, and (e) using dynamic-static layer skipping. The modules with light colour are not activated in current inference. We set the same legend for VLA layers in (a), (b), (c), (d), and dynamic layers in (e).

skipping all of these layers will cause extremely low accuracy. We define these layers as *dynamic layers* and dynamically skip them. Before each dynamic layer, we determine whether to perform layer skipping (the decision mechanism will be discussed in Section 3.3). If we decide to conduct layer skipping, we directly skip to the next static layer, which shows a higher speedup compared with skipping only one layer each time. And we train adapters (light-weight feedforward layers) to summarize the skipped layers and fit the activation for the next static layer. This is reasonable as the skipped layers will not change the activation distribution much and contain less information, which is within the adapter's fitting ability.

By using dynamic-static layer skipping, we can achieve a higher speedup with low information loss. At the same time, our method does not need to train multiple action heads or the LLM backbone, which reduces the training cost.

## 3.3 PRIOR-POST SKIPPING GUIDANCE

In this section, we discuss our strategy to determine whether to conduct layer skipping before each dynamic layer. As discussed in Section 3.1, just using skipping controllers Jiang et al. (2024); Luo et al. (2025) is neither accurate nor efficient, and we should keep enough layers for important actions. Based on this, we find that the trajectory continuity can reflect the importance of the current action. Here we define the continuity at step $t$ as:

$$C_t = -\frac{1}{k} \sum_{j=t-k+1}^{t} ||\delta A_j||_2 = -\frac{1}{k} \sum_{j=t-k+1}^{t} ||A_j - A_{j-1}||_2, \qquad (1)$$

where $\delta A_j$ represents the difference between action $j$ and $j-1$, and we consider the trajectory of last $k$ actions. As shown in Figure 6 (a) (b), we find that the trajectory has good continuity at most of the time, which means adjacent actions have similar magnitude and direction. This phenomenon comes from the training data collection process for VLA models, as robot operators tend to keep uniform speeds when executing non-critical motions. However, the continuity will be broken when the robots are conducting fine operations, e.g., grasping or releasing objects. Unlike free-space movements that follow smooth trajectories, these fine operations include frequent stops, micro-corrections, and hesitation, which breaks the natural flow of motion into disjointed segments Wang et al. (2024). We find that these actions show higher importance in task completion, and conducting layer skipping on these steps will introduce huge accuracy loss, as shown in Figure 6 (a) (b).

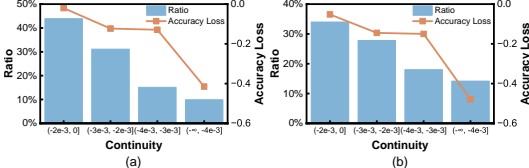

Figure 6: The proportion of action prediction steps in different continuity ranges and the accuracy loss when conducting layer skipping at the steps in different continuity ranges for (a) RoboFlamingo-3B and (b) RoboFlamingo-9B.

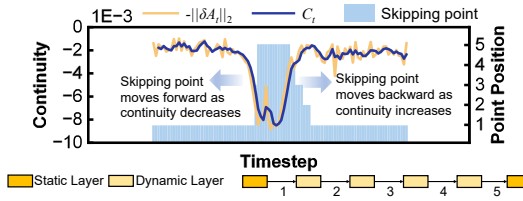

Figure 7: Pre-skip prediction and post-skip verification method. The modules with light colour are not activated in current inference.

Based on these observations, we can approximate action importance based on action continuity to guide layer skipping determination. As shown in Figure 7, besides skipping controllers, we propose pre-skip prediction to approximate the proper skipping positions between static layers. Between each two adjacent static layers, we define a skipping-allow point, which is initialized after the front static layer. Before the point, the skipping controllers are disabled, and the VLA layers are forcibly kept. The skipping controller after the point is activated and determines layer skipping. We dynamically move the skipping-allow point according to the continuity change of the previous $k$ steps:

$$l_i = l_i + \delta l, \quad \text{when } C_t - C_{t-1} < \eta_1 \text{ and } l_i < s_i, \tag{2}$$

$$l_i = l_i - 1, \quad \text{when } C_t - C_{t-1} > \eta_2 \text{ and } l_i > s_{i-1}, \tag{3}$$

where $l_i$ is the id of the layer after the skipping-allow point, $\eta_1$ and $\eta_2$ are thresholds ($\eta_1 < 0$), $s_i$ is the $i_{th}$ static layer, $\delta l$ is an adaptive moving stride defined as $\delta l = \lceil \frac{C_t - C_{t-1}}{\eta_1} \rceil$. Note that when detecting decreased continuity, the skipping allowance point rapidly shifts forward according to continuity change to prioritize action accuracy during critical phases. Conversely, when continuity improves, it gradually moves backward. This hysteresis-like system design maximizes the correctness of essential actions. And we decide the position of the skipping-allow points according to the trajectory of the previous $k$ steps instead of a single recent step ($k = 1$), as the change of action difference ($\delta A_t - \delta A_{t-1}$) will become low when important actions occur in a continuous mode. We will discuss the influence of $k$ value in Appendix C. In the first $k$ steps of a task, we do not move the skipping-allow points. An example of skipping-allow points moving is shown in Figure 8. Note that the pre-skip prediction only closes or activates the skip controllers, and whether to conduct layer skipping is still determined by the activated skip controller itself.

By using pre-skip prediction, we can keep enough layers for the important actions, thus improving model performance. At the same time, the extra latency cost caused by skipping controllers can also be reduced, as the pre-skip prediction will close most of the skipping controllers that will probably decide not to skip. In practice, we find that in each inference, usually only around 20% of the skipping controllers are truly used on average.

However, only using pre-skip prediction is not enough. This is because we can not get the current action prediction before current model inference, so the continuity decrease can only be

Figure 8: Skipping-allow point changes in the manipulation task. Here we use 4 dynamic layers between 2 static layers as an example, which has 5 possible positions for the skipping-allow point.

detected after the first important action has been predicted, whose correctness can not be guaranteed. The issue becomes more pronounced when the action chunk technique is used Kim et al. (2025); Black et al. (2024); Wen et al. (2025), where multiple actions will be predicted in a single inference. To solve this problem, we also propose post-skip verification to add a feedback mechanism, which is shown in Figure 7. When we first detect the continuity decrease ($\delta C_t = C_t - C_{t-1} < \eta_1$ and $\delta C_{t-1} = C_{t-1} - C_{t-2} > \eta_1$), we re-predict the current action without any layer skipping. Then we recompute the continuity change using the re-predicted action. This process not only ensures the correctness of the initial critical action prediction but also enhances the detection accuracy of continuity degradation. And it will not introduce much extra cost, as important actions only occupy a small proportion and usually appear continuously.

Figure 9: The (a) first stage and (b) second stage of skip-aware two-stage knowledge distillation method. The layers in red boxes are the selected layers in current training step.

## 3.4 SKIP-AWARE TWO-STAGE KNOWLEDGE DISTILLATION

As discussed in Section 2, previous VLA layer skipping works Yue et al. (2024); Zhang et al. (2025) require training the LLM backbone, which will introduce high training cost and generalization loss for VLA models. Our method freezes the LLM backbone and just trains the lightweight skipping controllers and adapters. This reduces training cost and ensures that our model can maintain the same accuracy as the original VLA when not using layer skipping. However, the training strategy is nontrivial, as simply training controllers and adapters together may cause a convergence problem. This is because the controllers and adapters are both randomly initialized, and the training of them will impact each other. At the beginning of training, as the adapters are not trained, the controllers will refuse layer skipping to avoid huge task loss. And this will further affect the adapter training.

To solve this problem, we propose skip-ware two-stage knowledge distillation. In the first stage, as shown in Figure 9 (a), we only train all the adapters to summarize the information of the following dynamic layers. The loss function of the first stage is:

$$loss_1 = \sum_i \|adapter_i(x_i) - L_{s_i-1}(L_{s_i-2}(\dots(L_i(x_i))))\|_F, \tag{4}$$

where $x_i$ is the input to dynamic layer $i$, $L_i$ is the $i_{th}$ layer and $s_i$ is the layer id of next static layer.

After the basic capability of adapters are developed, in the second stage, we can then train the controllers and adapters together, as shown in Figure 9 (b). However, in the forward path, if we use the controller itself to decide the skipping place, we find that the model will continuously skip at the same dynamic layer. So, to fully train each controller, between two static layers, we just select one dynamic layer $i$ ($s_{i-1} < i < s_i$) and predict the probability of skipping using the controller before this layer. In layer selection, it is important to select early dynamic layers more times, as the corresponding adapters are required to summarize more dynamic layers and need more training. In practice, we use Harmonic decay probability Bochner (2005) to select the dynamic layer, and we find that other strategies, such as linear decay probability, also work well. Different from the inference time, to make the skipping decision module differentiable, we conduct the forward propagation from layer $i$ to layer $s_i$ following:

$$x_{s_i} = controller_i(x_i) \cdot adapter_i(x_i) + (1 - controller_i(x_i)) \cdot L_{s_i-1}(L_{s_i-2}(\dots(L_i(x_i)))), \tag{5}$$

where $x_{s_i}$ is the activation input to the next static layer. We also introduce a normalization loss to encourage the controller to skip layers. The loss function of the second stage is:

$$loss_2 = task\_loss + \lambda \cdot \sum_i (1 - controller_i(x_i)) \cdot (s_i - i), \tag{6}$$

where $i$ belongs to the layer ids of the selected layers.

Although our training has two stages, its cost is far lower than previous layer skipping works, as we do not train the LLM backbone and need fewer training steps, which we show in Section 4.

## 4 EXPERIMENTS

### 4.1 EXPERIMENT SETUP

We evaluate DySL-VLA on CALVIN benchmark Mees et al. (2022b) using RoboFlamingo Li et al. (2023b) models and LIBERO benchmark Liu et al. (2023) using OpenVLA-oft models Kim et al.

Table 2: Accuracy comparison on Calvin D $\rightarrow$ D dataset.

| Method | # Fine-tuned Parameters | # Fine-tuned Steps | Training Cost (GPU Hour) | Task 1 | Task 2 | Task 3 | Task 4 | Task 5 | Avg Length | RTX 4090 Latency |
|---|---|---|---|---|---|---|---|---|---|---|
| HULC Mees et al. (2022a) | – | – | – | 0.827 | 0.649 | 0.504 | 0.385 | 0.283 | 2.64 | – |
| SPIL Zhou et al. (2024) | – | – | – | 0.846 | 0.651 | 0.508 | 0.380 | 0.286 | 2.67 | – |
| RoboFlamingo 3B Li et al. (2023b) | – | – | – | 0.871 | 0.696 | 0.496 | 0.371 | 0.272 | 2.71 | 51.0ms |
| RoboFlamingo 3B (Re-trained) | – | – | – | 0.903 | 0.694 | 0.486 | 0.403 | 0.333 | 2.82 | 51.0ms |
| DeeR-VLA 3B Yue et al. (2024) | 1.2B | $9.2 \cdot 10^4$ | 112 | 0.853 | 0.696 | 0.549 | 0.420 | 0.312 | 2.83 | 19.3ms |
| Random Skip | – | – | – | 0.322 | 0.045 | 0.011 | 0.000 | 0.000 | 0.38 | 22.6ms |
| FlexiDepth Luo et al. (2025) | 19M | $6.7 \cdot 10^3$ | 7 | 0.844 | 0.489 | 0.244 | 0.200 | 0.089 | 1.87 | 27.6ms |
| DySL-VLA 3B | 14M | $6.7 \cdot 10^3$ | 7 | 0.894 | 0.719 | 0.539 | 0.420 | 0.320 | **2.89** | 13.6ms |

Table 3: Accuracy comparison on LIBERO dataset.

| Method | # Fine-tuned Parameters | Spatial SR (%) | Object SR (%) | Goal SR (%) | Long SR (%) | Average SR (%) | A6000 Latency | Jetson Orin Latency |
|---|---|---|---|---|---|---|---|---|
| MDT (scratch) Reuss et al. (2024) | – | 78.5 | 87.5 | 73.5 | 64.8 | 76.1 | – | – |
| $\pi_0$ + FAST (fine-tuned) Pertsch et al. (2025) | – | 96.4 | 96.8 | 88.6 | 60.2 | 85.5 | – | – |
| OpenVLA-OFT 7B Kim et al. (2025) | – | 97.6 | 98.4 | 97.9 | 94.5 | 97.1 | 53.0ms | 676ms |
| DeeR-VLA 7B Yue et al. (2024) | 7.1B | 97.0 | 98.2 | 97.4 | 88.6 | 95.3 | 40.2ms | 495ms |
| Random Skip | – | 11.3 | 0.6 | 0.0 | 0.6 | 3.1 | 34.9ms | 383ms |
| FlexiDepth Luo et al. (2025) | 390M | 84.0 | 85.2 | 41.4 | 10.3 | 55.2 | 42.2ms | 502ms |
| DySL-VLA 7B | 226M | 98.0 | 98.2 | 97.0 | 92.6 | 96.5 | 27.4ms | 345ms |

Table 4: Individual influence of our methods (evaluated on Calvin D $\rightarrow$ D dataset).

| Method | Task 1 | Task 2 | Task 3 | Task 4 | Task 5 | Avg Length | Avg Latency |
|---|---|---|---|---|---|---|---|
| RoboFlamingo 3B Li et al. (2023b) | 0.903 | 0.694 | 0.486 | 0.403 | 0.333 | 2.82 | 51.0ms |
| Random Skip | 0.322 | 0.045 | 0.011 | 0.000 | 0.000 | 0.38 | 22.6ms |
| DySL-VLA | 0.894 | 0.719 | 0.539 | 0.420 | 0.320 | **2.89** | 13.6ms |
| w/o Post-skip Verification | 0.897 | 0.704 | 0.515 | 0.398 | 0.280 | 2.79 | 13.4ms |
| w/o Pre-skip Prediction | 0.870 | 0.635 | 0.426 | 0.296 | 0.192 | 2.42 | 20.1ms |
| w/o Skip-aware Two-stage Knowledge Distillation | 0.903 | 0.694 | 0.486 | 0.403 | 0.333 | 2.82 | 74.0ms |
| w/o Dynamic-static Layer Skipping | 0.844 | 0.489 | 0.244 | 0.200 | 0.089 | 1.87 | 27.6ms |

(2025). In the simulation platform, the robot can access RGBD observations. For Calvin datatset, the robot is instructed to complete a task sequence with five subtasks. Following Yue et al. (2024), model performance is evaluated based on the average successful length (0 to 5). For OpenVLA-oft, following Kim et al. (2025), we evaluate on 4 sub-datasets. More implementation details and the model architecture are shown in Appendix A. The rollout visualization of our method is shown in Appendix B. Our simulation-based evaluation is widely used in many prior works Kim et al. (2025); Yue et al. (2024); Li et al. (2023b), and our experimental setup follows the same standard, ensuring fair comparison. As our method mainly solves the problem of high computation cost, we also deploy our model on the computation platform (Jetson Orin) that is frequently used by real-world robots.

## 4.2 MAIN RESULTS

**Accuracy comparison.** The accuracy comparison is shown in Table 2 and 3. On Calvin D $\rightarrow$ D dataset, compared with traditional methods HULC and SPIL, which rely on hierarchical planning and skill priors, DySL-VLA shows accuracy improvement. This is because VLA models are trained on pre-trained VLMs embedded with internet-scale knowledge, thus showing higher generalization ability. Compared with FlexiDepth Luo et al. (2025), which only uses skipping controllers and adapters before each layer to conduct layer skipping, DySL-VLA shows 54.5% average successful length improvement, respectively, by using static-dynamic layer skipping to reduce information loss. DySL-VLA also shows 2.1% average successful length improvement over DeeR-VLA, with $85.7\times$ trainable parameters reduction and $13.7\times$ training steps reduction (further comparison is shown in Appendix E). Because DySL-VLA keeps the informative layers to avoid information loss, and uses pre-skip prediction and post-skip verification to ensure correction for important actions. Our method even shows successful length improvement over full RoboFlamingo model. We suggest that the original model trained on D $\rightarrow$ D dataset has redundant parameters and suffers from overfitting. Our method effectively eliminates these redundancies by layer skipping, resulting in a more robust architecture. For other Calvin datasets, we show the accuracy result in Appendix D. On LIBERO dataset, our method also shows 41.3% average SR improvement over FlexiDepth. Compared with DeeR-VLA, our method shows 1.2% average SR improvement, with $31.4\times$ trainable parameters reduction.

**Latency Comparison.** The average LLM latency comparison is shown in Table 2 and 3. On Calvin dataset, our method achieves $3.75\times$ latency reduction compared to the full RoboFlamingo model, by dynamically skipping unnecessary VLA layers. Our method achieves $2.03\times$ latency reduction compared with FlexiDepth Luo et al. (2025), which sets a skipping controller before each layer and only skips one layer each time, while our method can skip more layers each time by using static-dynamic layer skipping. And our method avoids redundant skipping controller inference by pre-skip prediction, which is further shown in Figure 10. Our method also shows $1.42\times$ latency reduction compared with early exit method DeeR-VLA Yue et al. (2024), with far less training cost. Because our method can keep the most informative layers of the original model, thus avoiding huge training costs to recover model performance and achieve high speedup. On LIBERO dataset, our method shows $1.54\times$and $1.47\times$ latency reduction on A6000 and $1.46\times$ and $1.43\times$ on Jetson Orin, compared with FlexiDepth and DeeR-VLA, respectively. Our method shows $1.93\times$ and $1.96\times$ latency reduction on A6000 and Jetson Orin compared with full RoboFlamingo model. This acceleration ratio is lower than RoboFlamingo, as OpenVLA-oft has lower parameter redundancy. Note that although deployment on Jetson Orin shows higher latency because of limited computation resources compared with A6000, the control frequency of DySL-VLA can still reach 23.2Hz, as OpenVLA-oft uses action chunk and predicts 8 actions in a single inference.

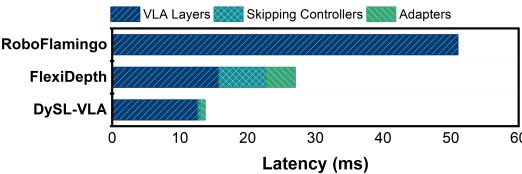

Figure 10: Latency breakdown of LLM backbone.

### 4.3 ABLATION STUDY

**Individual influence of our methods.** The individual influence of our methods is shown in Table 4. Not using pre-skip prediction and post-skip verification will both cause accuracy drop, as the correctness of important actions can not be fully guaranteed. Without dynamic-static layer skipping, the informative layers can not always be saved, resulting in accuracy drop. Without skip-aware two-stage knowledge distillation, the training of adapters and controllers will affect each other, and the controllers will always be closed, thus leading to higher latency, as discussed in Section 3.4.

**Impact of static layer ratio.** The impact of static layer ratio is shown in Table 5, evaluated on Calvin D $\rightarrow$ D dataset. Across various ratios, our method maintains both accuracy and latency within an acceptable range, demonstrating its robustness. Relatively, a lower static layer ratio reduces latency, but causes slight accuracy drop. For a higher ratio, the latency increases with little accuracy gain, as some less informative layers are kept as static layers. So a moderate ratio such as 20% is appropriate.

Table 5: Ablation study on static layer ratio.

| Static Layer Ratio | 10% | 15% | 20% | 25% | 30% |
|---|---|---|---|---|---|
| Average Length | 2.81 | 2.84 | 2.89 | 2.88 | 2.89 |
| Average Latency (ms) | 12.6 | 13.1 | 13.6 | 14.7 | 15.8 |

**Impact of skipping-allow point moving stride.** In pre-skip prediction, we use an adaptive moving stride based on continuity change when moving the skipping-allow point forward. Here we compare our strategy with constant strides, shown in Table 6. Our method overcomes all constant baselines, as the continuity can better reflect the action importance. We also find that a larger forward-moving stride shows relatively better results, as the hysteresis-like system design can better protect essential actions.

Table 6: Ablation study on $\delta l$ (evaluated on LIBERO Spatial dataset).

| $\delta l$ | 1 | 2 | 3 | 4 | 5 | Adaptive |
|---|---|---|---|---|---|---|
| SR | 96.2 | 96.4 | 96.4 | 97.4 | 97.0 | **98.0** |

### 5 CONCLUSION

In this paper, we propose DySL-VLA, accelerating VLA models by dynamically skipping unnecessary layers according to action importance. We propose dynamic-static layer skipping to statically keep the most informative layers and reduce information loss. We propose prior-post skipping guidance to guarantee we keep enough layers for important actions. We propose skip-aware two-stage knowledge distillation to improve training convergence. In experiments, DySL-VLA shows 2.1% average successful length improvement over DeeR-VLA on Calvin D $\rightarrow$ D dataset, with $85.7\times$ trainable parameters and $13.7\times$ training steps reduction.

## 6 ETHICS STATEMENT

The potential negative societal impacts of our method align with those typically associated with general robotic technologies. Fair and safe deployment principles in robotic systems are important.

## 7 REPRODUCIBILITY STATEMENT

To ensure the reproducibility of our work, our experimental setup is thoroughly detailed: we describe our dataset usage in Section 4.1, specify the model architectures used in Appendix A.1, and list the hyper-parameters (such as learning rates and batch sizes) for each experiment in Appendix A.2. The source code, which implements our core method and the main experiments, is available on Anonymous Github.

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

# A IMPLEMENTATION DETAILS

## A.1 NETWORK ARCHITECTURE

In this section, we introduce the model architecture of RoboFlamingo and OpenVLA-oft. The architectural details are shown in Table 7. RoboFlamingo is trained based on OpenFlamingo Alayrac et al. (2022), which is a VLM pre-trained using the web-scraped image-text datasets LAION-2B Schuhmann et al. (2022) and Multimodal C4 Zhu et al. (2023). OpenFlamingo uses MPT-1B Team et al. (2023) as its LLM backbone and introduces cross-attention layers before the original MPT layers to fuse the information from image and text. RoboFlamingo uses an LSTM model Yu et al. (2019) followed with Multi-layer Perceptron (MLP) head as the action head. Given the output activation from the LLM backbone, the action head predicts a continuous action with 7 dimensions. OpenVLA-oft is trained based on LLaMA 7B, with SigLIP and DINO V2 as the vision encoder. Different from RoboFlamingo, OpenVLA-oft uses action chunk technique. OpenVLA-oft predicts 8 actions in each inference, and each action has 7 dimensions. The input of OpenVLA-oft LLM backbone includes vision tokens, robot state tokens, and several empty action tokens. After LLM inference, the MLP action head takes the output feature of the action tokens and predicts continuous actions. In this paper, we focus on accelerating the LLM backbone of the VLA model.

## A.2 TRAINING DETAILS

In our method, we only train the light-weight adapters and the skipping controllers, instead of the LLM backbone, which largely reduces the training cost. And we propose skip-aware two-stage knowledge distillation to improve model convergence. We report our training hyper-parameters in Table 8, 9 and Table 10.

Table 7: Model architectural details.

| Model | LLM Backbone | Vision Encoder | # LLM Layers | Cross-attention Interval | Hidden Dimension | Action Head |
|---|---|---|---|---|---|---|
| OpenFlamingo 3B | MPT-1B (Instruct) Team et al. (2023) | CLIP ViT-L/14 | 24 | 1 | 2048 | LSTM+MLP |
| OpenVLA-oft 7B | LLaMA 7B | SigLIP + DINO v2 | 32 | – | 4096 | MLP |

Table 8: Training hyper-parameters for RoboFlamingo (D → D dataset).

| Hyper-parameters | Values |
|---|---|
| Batch size | 24 |
| Optimizer | AdamW |
| learning rate | $1 \times 10^{-4}$ |
| Learning rate schedule | constant |
| # steps of the first training stage | 1030 |
| # steps of the second training stage | 5690 |
| $\lambda$ | 5e-4 |
| LSTM window size | 12 |

Table 9: Training hyper-parameters for RoboFlamingo (ABC → D dataset).

| Hyper-parameters | Values |
|---|---|
| Batch size | 24 |
| Optimizer | AdamW |
| learning rate | $1 \times 10^{-4}$ |
| Learning rate schedule | constant |
| # steps of the first training stage | 3570 |
| # steps of the second training stage | 12780 |
| $\lambda$ | 5e-4 |
| LSTM window size | 12 |

Table 10: Training hyper-parameters for OpenVLA-oft.

| Hyper-parameters | Values |
|---|---|
| Batch size | 32 |
| Optimizer | AdamW |
| learning rate | $1 \times 10^{-4}$ |
| Learning rate schedule | constant |
| # steps of the first training stage | 7693 |
| # steps of the second training stage | 29307 |
| $\lambda$ | 1e-2 |

## B  ROLLOUT VISUALIZATION

We show the visualization of our method in Figure 11.

Task 1: Press the button to turn off the led light.

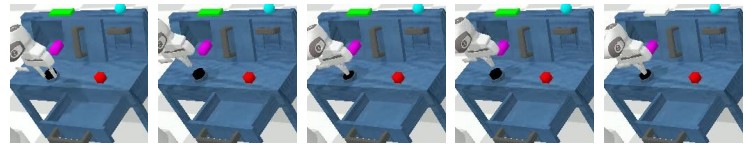

Task 2: Slide the block that it falls into the drawer.

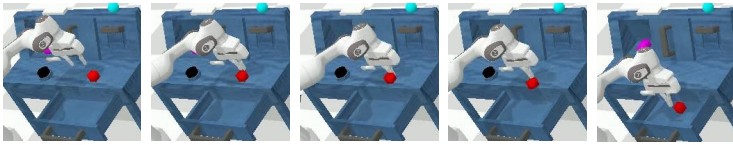

Task 3: Pull the handle to open the drawer.

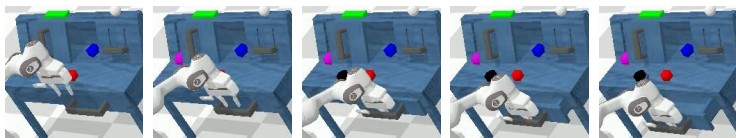

Task 4: Grasp, lift the pink block and store it in the sliding.

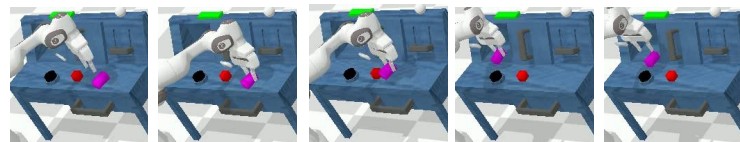

Task 5: Take the blue block and rotate it to the right.

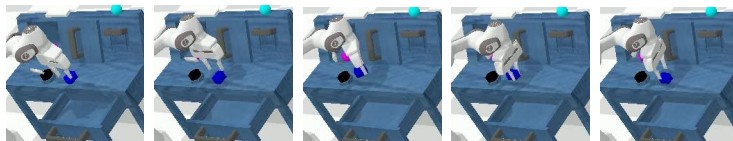

Figure 11: Visualization of DySL in 5 manipulation tasks.

## C  INFLUENCE OF THE TRAJECTORY LENGTH IN PRE-SKIP PREDICTION

In pre-skip prediction, we consider the trajectory continuity of previous $k$ actions. The influence of $k$ value is shown in Figure 12. In our experiment, we set $k = 5$, which shows good model performance. And there will be model performance drops when $k$ is too large or too small. When $k$ is too small, the short trajectory can not reflect the action importance well. And a large $k$ will reduce the sensitivity to continuity changes.

Table 11: Accuracy comparison on Calvin ABC → D dataset.

| Method | # Fine-tuned Parameters | # Fine-tuned Steps | Training Cost (GPU Hour) | Task 1 | Task 2 | Task 3 | Task 4 | Task 5 | Avg Length |
|---|---|---|---|---|---|---|---|---|---|
| HULC Mees et al. (2022a) | – | – | – | 0.418 | 0.165 | 0.057 | 0.019 | 0.011 | 0.67 |
| SPIL Zhou et al. (2024) | – | – | – | 0.742 | 0.463 | 0.276 | 0.147 | 0.080 | 1.71 |
| RoboFlamingo 3B Li et al. (2023b) | – | – | – | 0.859 | 0.674 | 0.487 | 0.317 | 0.025 | 2.59 |
| RoboFlamingo 3B (Re-trained) | – | – | – | 0.861 | 0.697 | 0.528 | 0.428 | 0.329 | 2.85 |
| DeeR-VLA 3B Yue et al. (2024) | 1.2B | $1.8 \cdot 10^5$ | 192 | 0.862 | 0.701 | 0.518 | 0.415 | 0.304 | 2.82 |
| Random Skip | – | – | – | 0.385 | 0.060 | 0.009 | 0.000 | 0.000 | 0.45 |
| FlexiDepth Luo et al. (2025) | 19M | $1.6 \cdot 10^4$ | 15 | 0.720 | 0.452 | 0.238 | 0.154 | 0.090 | 1.65 |
| DySL-VLA 3B | 14M | $1.6 \cdot 10^4$ | 15 | 0.868 | 0.711 | 0.525 | 0.419 | 0.303 | **2.83** |

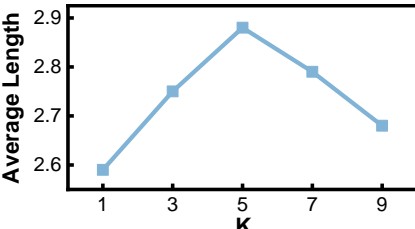

Figure 12: Influence of the value of $k$ in pre-skip prediction, evaluated on Calvin D → D dataset.

# D    ACCURACY ON CALVIN ABC → D DATASET.

The accuracy comparison on Calvin ABC → D dataset is shown in Table 11. Compared with traditional methods HULC Mees et al. (2022a) and SPIL Zhou et al. (2024), which rely on hierarchical planning and skill priors, DySL-VLA shows higher generalization ability and large accuracy improvement. This is because VLA models are trained on pre-trained VLMs, which is embedded with internet-scale knowledge. Compared with FlexiDepth Luo et al. (2025), which uses skipping controllers and adapters before each layer to conduct layer skipping, DySL-VLA shows 71.5% average successful length improvement, respectively, by using static-dynamic layer skipping to reduce information loss. DySL-VLA shows similar successful length compared with DeeR-VLA Yue et al. (2024), with 85.7× trainable parameters reduction and 11.3× training steps reduction. This is because DySL-VLA keeps the informative layers to avoid large-scale training, and uses pre-skip prediction and post-skip verification to ensure correction for important actions.

# E    FURTHER COMPARISON WITH DEER-VLA.

Here we further compare our method with the most relevant baseline DeeR-VLA Yue et al. (2024), as shown in Table 12. DeeR-VLA skips all the final layers when conducting early exit, which may cause huge information loss. So it re-trains the LLM backbone to recover model performance, while large-scale fine-tuning on specific scenarios may break the generalization ability of VLA models and introduce huge training costs. We further evaluate DeeR-VLA with the LLM backbone frozen (only training multiple action heads), which leads to performance drop. In contrast, our method use dynamic-static layer skipping to keep the most informative layers, which prevents training the LLM backbone and only fine-tunes the light-weight adapters and skipping controllers.

Table 12: Further comparison with Deer-VLA (evaluated on Calvin D → D dataset).

| Method | # Fine-tuned Parameters | # Fine-tuned Steps | Avg Length |
|---|---|---|---|
| RoboFlamingo 3B Li et al. (2023b) | – | – | 2.82 |
| DeeR-VLA 3B Yue et al. (2024) | 1.2B | $9.2 \cdot 10^4$ | 2.83 |
| DeeR-VLA 3B Yue et al. (2024) (LLM backbone frozen) | 14M | $6.7 \cdot 10^3$ | 1.95 |
| DySL-VLA 3B | 14M | $6.7 \cdot 10^3$ | **2.89** |

