# OpenReview forum: "DySL-VLA: Efficient Vision-Language-Action Model Inference via Dynamic-Static Layer-Skipping for Robot Manipulation"
_ICLR.cc/2026/Conference — ICLR 2026 Conference Withdrawn Submission_

### Official Review · Reviewer_aNrR · 2025-10-28

**Soundness:** 3
**Presentation:** 2
**Contribution:** 2
**Rating:** 2
**Confidence:** 4

**Summary:**

This paper introduces DySL-VLA, a framework that improves the efficiency of VLA model inference by dynamically skipping model layers based on the importance of robot actions. The method distinguishes between static (informative) and dynamic (skippable) layers and leverages trajectory continuity as an indicator of action importance. Through dynamic-static layer skipping, pre- and post-skip guidance, and skip-aware two-stage knowledge distillation, the model achieves acceleration with minimal loss in accuracy. Experiments on the CALVIN and LIBERO benchmarks show that DySL-VLA improves inference speed and even slightly enhances performance over existing baselines.

**Strengths:**

1.	The paper leverages the continuity of the trajectory as an indicator of the importance of current actions, which is intuitive and aligns with robotic motion characteristics.
2.	The proposed dynamic-static layer skipping is a well-motivated and technically sound idea to balance accuracy and efficiency by preserving informative layers while skipping redundant ones.

**Weaknesses:**

1.	The evaluation falls short of demonstrating real-world applicability. Given the claim of improved efficiency for robotic deployment, a real-robot experiment on edge hardware (e.g., Jetson Orin) with latency and performance measurements would strongly support the paper’s claims. The current evaluation is limited to simulations and lacks evidence of generalization to diverse real-world scenarios.
2.	Several method components rely on heuristics that may limit generality. For instance, defining static layers by average cosine similarity or selecting thresholds ($\eta_1, \eta_2$) and the number of static layers to keep are not clearly justified or analyzed for transferability across models or tasks.
3.	The definition of “important actions” could be better clarified. While the link between low trajectory continuity and higher action importance is plausible, it is not rigorously demonstrated. Small errors in supposedly “unimportant” actions can still accumulate and degrade performance, so the assumption that such actions are safely skippable requires more discussion.
4.	Continuity may serve as a useful heuristic for identifying important actions but may not be sufficient on its own. Other factors, such as task semantics or environmental variability, might also influence action importance.

Minor Points:

1.	Please use **\citep** properly instead of **\cite** .
2.	Use proper quotation marks (**``''** ) in LaTeX.

**Questions:**

1. The paper notes: “However, in the forward path, if we use the controller itself to decide the skipping place, we find that the model will continuously skip at the same dynamic layer.” Could the authors explain why this occurs and why it is undesirable?
2.	In the evaluation, could the authors clarify why performance improves when layers are skipped? Is the improvement due to reduced overfitting or more effective training under the two-stage strategy?

---

### Official Review · Reviewer_zimE · 2025-10-28

**Soundness:** 3
**Presentation:** 3
**Contribution:** 3
**Rating:** 6
**Confidence:** 2

**Summary:**

The paper introduces three mechanisms to improve inference latency and reduce the number of trainable parameters in vision-language-action (VLA) models. First, model layers are partitioned into static and dynamic groups based on their empirically determined importance. Second, an adaptive algorithm uses the continuity of predicted actions computed from a running average to ensure enough layers are kept during critical moments. Third, a two-stage training procedure stabilizes learning: adapters are first trained to summarize skipped layers, then controllers and adapters are jointly optimized using skip-aware knowledge distillation, enabling efficient layer skipping without degrading task performance.

**Strengths:**

- The paper is well written and presents extensive empirical evidence, such as activation similarity and layer significance in VLA settings to support the investigation of adaptive layer skipping.
- To the best of my knowledge, the introduced algorithms for pre-skip prediction and two-stage knowledge distillation are novel in the context of VLA training. Both mechanisms are clearly motivated, theoretically sound, and directly address the challenges of inference speed, parameter efficiency, and maintaining high success rates.
- The results on the LIBERO benchmark are strong, with DySL-VLA achieving competitive success rates while reducing the number of trainable parameters by over 30x compared to the state-of-the-art OpenVLA-OFT model.
- By applying the method to two base models (OpenVLA-OFT and RoboFlamingo) across two different benchmarks (LIBERO and Calvin), the generality of the proposed approach is well supported.

**Weaknesses:**

- The method seems slightly convoluted and introduces several interacting components which come with their own set of hyperparameters respectively (e.g. static layer selection, continuity thresholds, trajectory window for continuity calculation, adapter architecture, controller thresholds, moving stride and number of training steps for each stage). While it was demonstrated to work on two different base VLA architectures and benchmarks, I am doubtful about the ease of adaptability to new tasks or architectures.
- The experiments on Calvin are only evaluated on the D->D and ABC->D settings, where performance of the base RoboFlamingo model is generally low. It remains unclear if DySL-VLA still performs well in the ABCD->D setting, where RoboFlamingo performs more consistently.

**Questions:**

- Can you elaborate on how static layers are chosen? The process seems data-driven, but I could not find an exact procedure to determine them.
- Could you comment on the general sensitivity of hyperparameters aside from those already mentioned in the ablation studies?
- How does the number of trainable parameters compare to e.g. LoRA finetuning?

---

### Official Review · Reviewer_uX9j · 2025-10-31

**Soundness:** 2
**Presentation:** 3
**Contribution:** 2
**Rating:** 4
**Confidence:** 3

**Summary:**

The paper presents DySL-VLA, a framework for speeding up VLAs dynamically skipping layers during inference. The method uses a hand-crafted continuity metric to decide when to skip, and lightweight adapters/controllers to maintain compatibility between layers. Experiments on simulation benchmarks show moderate speedups.

**Strengths:**

* **Motivation**: the approach tackles the interesting problem of improving VLA model latency and computational cost, which is relevant for robotics.
* **Latency improvements**: Provides speedup and some accuracy improvements, with ablation studies and reproducibility details.
* **Presentation** : the work is clearly presented and motivated. The observations and analysis in Section 3.1 provide useful insights into previous approaches and the proposed method.

**Weaknesses:**

* **Generalizability**: The decision to skip layers is based on a non-learned continuity calculation from action outputs, which may not generalize or be optimal.
* **Worsens performance?**: while the authors show an improved latency of the model, it looks like the proposed skipping actually damages performance. It is only a 1% reduction on LIBERO, but it is unclear what's the impact on CALVIN, as the authors don't report OpenVLA-OFT number (base VLA model adopted) in the Calvin table.

**Questions:**

* What is OpenVLA-OFT performance on CALVIN?
* State-of-the-art claims should be removed. OpenVLA-OFT, which the method is based on, is not shown for CALVIN. Furthermore, with the constantly evolving field, the values reported are already surpassed by many other approaches, e.g. SEER [1]


[1] Tian et al, Predictive Inverse Dynamics Models Are Scalable Learners for Robotic Manipulation

---

### Official Review · Reviewer_Q8hS · 2025-11-01

**Soundness:** 3
**Presentation:** 2
**Contribution:** 3
**Rating:** 6
**Confidence:** 3

**Summary:**

The paper presents an algorithm to speed up VLA inference by skipping layers in the VLA architecture. The paper posits that some layers in the VLA are more important than the other and should be kept in the architecture at all times whereas other layers may be skipped. The paper also observes that some actions are more “important” than others and the VLA shows high sensitivity to skipping those important actions and therefore presents a “prior-post skipping guidance mechanism” to determine when to skip those layers. To compensate for the skipped layers, the method trains adapters using a 2 staged knowledge distillation algorithm.

**Strengths:**

1. The paper presents a simple algorithm for layer skipping based on an observation that some layers in the VLA are much more important than others and some actions during the trajectory are important and require being processed by the full VLA whereas other actions are more amenable to layer skipping.
2. The method shows good performance on 2 benchmarks on 2 different VLA architectures.
3. The paper also presents extensive ablations on every part of the method showing individual performance improvements.

**Weaknesses:**

***1. Details on the thresholds***

How are $\eta_1$ and $\eta_2$ computed? If they are hyper-parameters, how do they change between models and simulators / tasks?

In general, the paper lacks a bit of detail on how do they come up with the hyper-parameters used in the experiments and it would be nice to include that.

***2.Real world experiments?***

The paper does not show any real-world performance. It would be nice to see how it affects the inference speed in the real-world.


***3. Writing.***

The writing in this paper can be much improved. Overall, the paper reads well and main method is well motivated but some parts of the text are weirdly phrased which hurt the readability. Noting some of those instances below (there might be more):
A. L147: “we conduct the following observations”? -> maybe rephrase to “we observe”
B. L150: “which layers we should skip to” -> which layers to skip ?
C. L198: very weird phrasing. Maybe simplify by just saying dynamically skip unnecessary layers.
D. L285: “layers are forcibly kept”
E. most of the quotation marks are wrong.

**Questions:**

Please see the weaknesses section for questions.
In general, I like the paper as it starts with an interesting observation and comes up with a simple fix to layer skipping leading to good performance improvements. I’m leaning towards a weak accept rating.

---

### Note · Authors · 2025-11-18

I have read and agree with the venue's withdrawal policy on behalf of myself and my co-authors.